# Seeing in Words: Learning to Classify through Language Bottlenecks

**Khalid Saifullah[1], Yuxin Wen[1], Jonas Geiping[1], Micah Goldblum[2], and Tom Goldstein[1]**
[1]University of Maryland, [2]New York University
{khalids, ywen, jgeiping, tomg}@umd.edu, goldblum@nyu.edu

## Abstract

Neural networks for computer vision extract uninterpretable features despite achieving high accuracy on benchmarks. In contrast, humans can explain their predictions using succinct and intuitive descriptions. To incorporate explainability into neural networks, we train a vision model whose feature representations are text. We show that such a model can effectively classify ImageNet images, and we discuss the challenges we encountered when training it.

## 1 Introduction

In recent years, there has been a surge of interest in vision-language models (VLMs) that combine the power of computer vision and natural language processing to perform tasks such as image captioning, visual question answering, and image retrieval (Alayrac et al., 2022; Radford et al., 2021; Li et al., 2022b; Wang et al., 2022; Zeng et al., 2021; Singh et al., 2022). These models leverage both visual and textual signals to reason about their contents and generate meaningful outputs (Li et al., 2022a; Xu et al., 2015; Anderson et al., 2018; Li et al., 2019; Zhou et al., 2020; Li et al., 2020).

One popular approach to building VLMs is through self-supervised learning (SSL), which involves training a model to make predictions about a given input without any human-labeled annotations. SSL has shown great promise in achieving state-of-the-art performance on various tasks in computer vision and natural language processing (Balestriero et al., 2023; Devlin et al., 2018).

Prior work has explored generating text descriptions from images using a variety of approaches. (Liu et al., 2023) encode images into text tokens using a pretrained codebook, but their generated text may lack semantic meaning. (Wickramanayake et al., 2021) use CNNs to associate visual features with human-annotated word phrases but require a manual definition of those phrases. Our work differs by employing an image-grounded language model decoder, eliminating the need for a codebook. This allows us to generate text descriptions that are semantically meaningful without relying on predefined word phrases.

In this paper, we take a stab at implanting a language bottleneck in traditional image classification pipelines (see Figure 1). By converting image features into words and using the words to classify the image, our proposed method can provide insights into the interpretability of classification models, as the language bottleneck serves as a "universal interface" between the visual and textual modalities. Extracting human-readable language features can also help us better understand how these models learn and reason about the content of images.

## 2 Method

We use BLIP (Li et al., 2022b), a fine-tuned image-to-text caption model as the basis of our pipeline. The model has two modules, a visual transformer (Dosovitskiy et al., 2020), which transforms input images into embedding vectors, and a language model (Devlin et al., 2018) that generates hard tokens by incorporating the signals from image embeddings with the help of the cross-attention layer.

To ensure that the pipeline remains end-to-end and differentiable, we feed $n$ trainable soft prompts as gold input to the text encoder instead of sampling to generate hard tokens. After that, the decoder

Table 1: Validation Results on ImageNet.

| Method | ImageNet | +Gaussian | +Impulse | +Shot | +Defocus |
|---|---|---|---|---|---|
| BLIP Caption | 42.819 | 40.185 | 38.874 | 39.866 | 39.147 |
| Ours | 67.117 | 64.799 | 63.201 | 64.784 | 64.287 |
| + Token similarity | 68.894 | 66.444 | 64.961 | 66.466 | 65.774 |
| + LLM loss | 64.035 | 62.162 | 60.89 | 62.195 | 61.602 |
| + No repetition sampling | 62.050 | 59.935 | 58.39 | 59.952 | 59.721 |

produces $n$ logits, we normalize them with softmax and perform a matrix-matrix multiplication with the word embedding matrix. This multiplication results in $n$ word embeddings for the image. To obtain a single vector, we mean pool the $n$ word embedding vectors. Finally, we pass the pooled vector through a linear classification head to predict the class.

Note that we only train the soft prompt and the linear head parameters. Also, during validation, we use argmax on the logits to retrieve the word embeddings of hard tokens. As a result, the linear head only "sees the words" to make its predictions.

Training such a model yields a challenging optimization problem and often leads to a model that mostly outputs non-human-readable text and repeated words. Therefore, we also design three variants to produce more diverse and human-readable image descriptions:

**Token similarity loss:** We compute the cosine similarity between all token pairs in the sequence, then calculate the average to get the sequence word similarity. This number tells us how much the generated tokens are similar to each other, and we minimize this loss to get more diverse tokens in the sequence.

**LLM loss:** In order to get human-readable text, we feed the generated tokens that we get from our language bottleneck through the language model again to get the likelihood of those tokens (bad language text should produce low likelihood).

**No repetition sampling:** This is a sampling procedure we use during inference, where we perform auto-regressive sampling but skip the tokens that were already generated in the sequence.

## 3 RESULTS

We test our pipeline on ImageNet (Deng et al., 2009) as well as ImageNet test sets with common corruptions (Hendrycks & Dietterich, 2019). We train the soft prompt and linear head for 5 epochs with learning rate $1e^{-1}$ and $5e^{-3}$ respectively. A natural candidate for a baseline is a system that uses the caption from the fine-tuned BLIP model, the motivation behind this is that BLIP has learned through the supervision of human captions on images, so it describes images the way a human might. But as the results show in Table 1, we do not get the optimal classification results with this baseline. Our method substantially improves the baseline accuracy. Meanwhile, adding token similarity loss produces the best validation accuracy. From our manual evaluation, training with token similarity loss drives the model to produce text tokens that have the best balance of being human-readable and helpful for the classifier. We provide sample generations in Appendix Figure 2.

## 4 CONCLUSIONS

In this paper, we proposed a method for implanting a language bottleneck in traditional image classification pipelines and investigated the performance of such a model. We found that incorporating language into the pipeline produces non-trivial classification accuracy on the ImageNet dataset. Our results suggest that language can serve as a universal interface and models can learn to express visual features through words alone. This kind of pipeline may provide insights into the interpretability and performance of vision-language models. Future work might explore better ways to make the language bottleneck produce more salient and human-readable words, and use this kind of pipeline to experiment on out-of-domain image samples. It would also be valuable to investigate how this approach can handle more complex images with multiple objects or scenes, and to consider its implications for model transparency and accountability.

## 5 ACKNOWLEDGEMENTS

This work was made possible by the ONR MURI program, DARPA GARD (HR00112020007), the Office of Naval Research (N000142112557), and the AFOSR MURI program. Commercial support was provided by Capital One Bank, the Amazon Research Award program, and Open Philanthropy. Further support was provided by the National Science Foundation (IIS-2212182), and by the NSF TRAILS Institute (2229885).

### URM STATEMENT

The authors acknowledge that at least one key author of this work meets the URM criteria of ICLR 2023 Tiny Papers Track.

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

# A APPENDIX

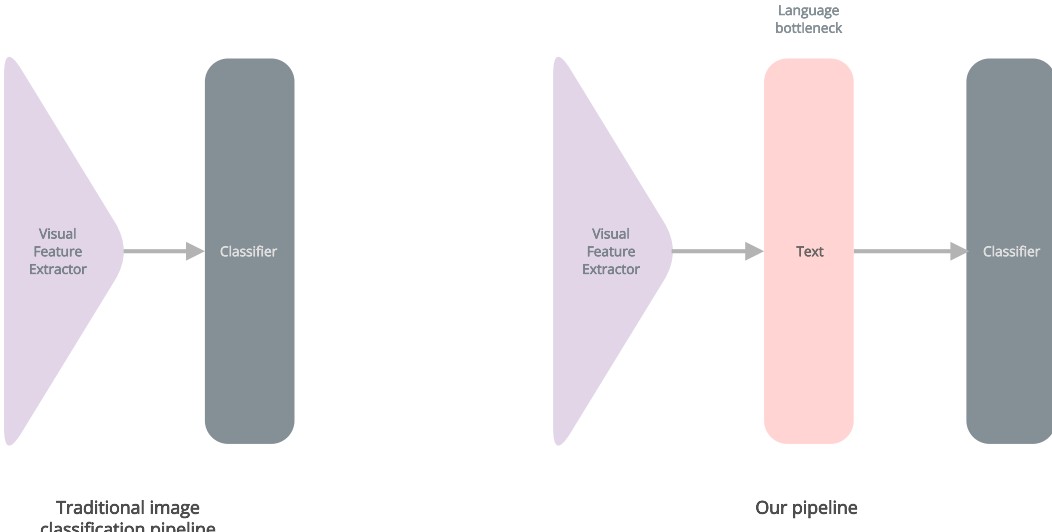

Figure 1: Architecture

| Image | BLIP Caption | Ours | +With No Repetition | Label |
|---|---|---|---|---|
|  | a picture of there are three people standing together in a lab | three fish oyster aquarium aquarium oyster oyster oyster | three a seafood oyster shell table aba tray bowl | lab coat |
|  | a picture of a bunch of kids sitting down in front of a bass | there musician trombone trombone trombone | there a music trombone trumpets chair brass trumpet | trombone |
|  | a picture of a dog is sitting in the grass with a frisbee | there dog span span bern bern bern bern bern | there a black english span dog border hound puppy | English springer |
|  | a picture of a man in military gear using a machine gun | rifle gun rifle rifle rifle rifle rifle rifle rifle | rifle a military army ak gun ar machine man | rifle |
|  | a picture of a totemologist is sitting on the back of a | there bird to to statue to to to pole | there a native to hai pole thunder statue hook | totem pole |

Figure 2: Sample Generations

