# OpenReview forum: "Seeing in Words: Learning to Classify through Language Bottlenecks"
_ICLR.cc/2023/TinyPapers — Submitted to Tiny Papers @ ICLR 2023_

### Official Review · Reviewer_r81c · 2023-03-18

**Confidence:** 4

**Summary Of Contributions:**

The paper proposes a novel approach to incorporating explainability into neural networks for computer vision.

**Rating:**

Clear, Correct, and Reproducible (CCR): a submission which meets the reviewing criteria

**Strengths And Weaknesses:**

The paper proposes a novel approach to incorporating explainability into neural networks for computer vision. The authors use a vision model whose feature representations are text to classify ImageNet images. They argue that current computer vision models extract uninterpretable features despite achieving high accuracy on benchmarks, whereas humans can explain their predictions using succinct and intuitive descriptions.

Summary Of Strengths:
* The paper addresses an important problem in computer vision: the lack of interpretability of current models.
* The proposed approach provides a way to obtain human-readable explanations of the classification decisions, which can be useful for applications where transparency and accountability are important.
* The model achieves high accuracy on ImageNet while also providing text descriptions that are consistent with the visual content of the images.

Weaknesses
* The proposed approach requires large amounts of textual annotations, which can be expensive and time-consuming to obtain. This may limit the applicability of the method to domains where such annotations are readily available.
* The paper does not provide a thorough analysis of the limitations of the proposed approach. For example, it is unclear how well the method would perform on images that contain multiple objects or scenes, or on tasks that require more complex reasoning.




**Suggested Changes:**

* The paper would benefit from a more detailed discussion of the ethical implications of incorporating interpretability into computer vision models, especially in domains where the decisions made by the models can have significant consequences.
* There are some minor typos and errors in the paper, such as missing words and inconsistent capitalization. These should be corrected in a revised version of the paper.

---

### Official Review · Reviewer_qB4U · 2023-04-02

**Confidence:** 5

**Summary Of Contributions:**

This paper proposes a new image classification model that can explain its decision in natural language and provide the classification decision. The proposed model is tested using the ImageNet datasets and has achieved good results.

**Rating:**

High Potential (HP): a submission which meets the reviewing criteria and has potential to make an impact on the field

**Strengths And Weaknesses:**

Strengths of the paper

* Providing explanations in natural language is an interesting approach to improving the human understandability of Deep Learning models. Hence, this paper addresses an interesting problem.

* The authors have proposed a simple and intuitive method to achieve the above objective.

* Overall, the paper is well-written, conveying the key message.

* The authors have backed up their claims with experiments using ImageNet.

* Since the authors have mentioned how they have built their models and given the respective references, someone interested would be able to reproduce the results indicated in the paper.

* The paper follows the basic requirements of the conference.

Weaknesses of the paper

*While the concept brought up by the paper is new, this is not the first paper to propose incorporating textual information to improve the interpretability of image classifiers and provide explanations in natural language. For example, authors are encouraged to check the CCNN proposed in [1] and highlight how their work differs from [1].

[1]  S. Wickramanayake, W. Hsu and M. L. Lee, "Comprehensible Convolutional Neural Networks via Guided Concept Learning," 2021 IJCNN.

*  Another main concern about this paper is the generated explanations merely describe the image content but do not highlight the visual concepts that could have influenced the model. For example, for an image to be classified as English Springer, the model should focus on specific features of the dog, especially because there is more than one type of dog in ImageNet. The main reason for this could be that the BLIP model is trained with merely the image descriptions, and that description does not particularly describe the objects in the image that correspond to the respective ImageNet class. Instead of ImageNet, authors can try their method with the CUB dataset, as it contains text descriptions describing the bird species.

* Some of the generated textual explanations need to be grammatically and semantically correct.

* Another problem is that despite using additional text information (even though the text information is not directly used, BLIP is trained using text data), the model's accuracy is not on par with the state-of-the-art classifiers on ImageNet. If authors can improve classification accuracy, the paper's contribution can be improved significantly.

**Suggested Changes:**

Please check the strengths and weaknesses section.

---

### Author Response · Authors · 2023-05-31
**Response to ICLR 2023 TinyPapers Paper146 Reviewers qB4U and r81c**

Thank you to both reviewers for taking the time to review our paper. We appreciate your valuable feedback and suggestions. We have carefully considered your comments and have made the necessary revisions to address the concerns raised. Below, we provide a detailed response to each reviewer's comments:

To Reviewer 1 (qB4U): We appreciate you highlighting the related work by Wickramanayake et al. We have now included a discussion of their CCNN model in the introduction section and clarified how our approach differs. Specifically, whereas CCNN uses human-defined word phrases to guide the learning of CNN features, our model directly uses an image-grounded language model decoder, allowing it to produce natural language explanations. We agree that the quality of the generated explanations can be improved, and we will also experiment with datasets that provide richer image annotations, like CUB, to help the model learn these concepts.

To Reviewer 2 (r81c): Thank you for the positive feedback and recognition of our work's contributions. We agree that our approach requires abundant textual annotations which may limit its applicability. In the conclusion section, we have now provided a discussion of potential limitations, including challenges related to handling complex scenes with multiple objects. Additionally, we emphasize the importance of future work exploring issues surrounding model transparency and accountability. Furthermore, we have diligently revised the paper to rectify the typos and inconsistencies you kindly pointed out.

Please let us know if you have any other questions or require clarification on any part of the revised submission.

---

### Meta-Review · Area_Chair_K58T · 2023-04-07

**Recommendation:** Invite to archive
**Confidence:** 4

**Metareview:**

This work proposes a new image classification model that can explain its decision in natural language and achieved good results on the ImageNet dataset. The strengths of the paper are the interesting approach to improving the human understandability of Deep Learning models, the simple and intuitive method proposed, and the well-written paper with backed-up claims through experiments.The weaknesses are that similar concepts have been previously proposed, generated textual explanations need to be improved, and the model's accuracy is not on par with state-of-the-art classifiers. Overall, the paper has high potential.

**Summary:**

This paper proposes a new image classification model that can explain its decision in natural language.

**Comments And Feedback To The Authors:**

- The authors should check similar concepts previously proposed and highlight how their work differs.
- Generated textual explanations need to be improved to describe visual concepts that influenced the model's decision better.
- Using additional text information, the model's accuracy is not on par with state-of-the-art classifiers. Improving classification accuracy can improve the paper's contribution significantly.


**Reason For Not Giving A Higher Recommendation:**

The reviewers note some weaknesses such as similar concepts previously proposed, the need for improvement in generated textual explanations, and the model's lower accuracy compared to state-of-the-art classifiers.

**Reason For Not Giving A Lower Recommendation:**

The reviewers' assessments are in agreement and suggest that the paper meets the reviewing criteria with high potential.

---

### Decision · Program_Chairs · 2023-04-08

Invite to archive